

# Analysis of technical characteristics of typical lower limb balance movements in Tai Chi: a cross-sectional study based on AnyBody bone muscle modeling

Haojie Li[1,*], Xin Wang[2,*], Zhihao Du[3] and Shunze Shen[4]

[1] School of P.E and Sports, Beijing Normal University, Beijing, China
[2] Zhengzhou University, Zhengzhou, China
[3] China University of Mining and Technology, Xuzhou, China
[4] Southwest Jiaotong University, Chengdu, China
[*] These authors contributed equally to this work.

Corresponding author
Xin Wang, 2264727485@qq.com

## ABSTRACT

**Background**. Tai Chi is an excellent traditional Chinese physical fitness exercise, and its unique balancing movements are very important for improving human balance. In this study, the two most representative Tai Chi balance movements, "knee lift balance" and "leg stirrup balance", were selected to establish the lower limb bone muscle model of Tai Chi balance movements by using computer simulation modeling technology, aiming to analyze the characteristics of the lower limb movement mechanisms of typical balance movements, to provide a quantitative theoretical basis for improving the scientific level of Tai Chi.

**Method**. A total of 16 subjects were recruited for this study. the BTS three-dimensional motion capture system and three-dimensional force platform were used for motion data acquisition, the physiological electromyographic signals were collected using BTS surface electromyography, and the lower limb bone muscle model of Tai Chi balance movements was established by AnyBody human simulation.

**Result**. In the knee lift balancing movement, the balance leg hip abduction/adduction angle, hip flexion/extension moment, and the strength of the rectus femoris muscle, biceps femoris short capitis, and iliacus muscle of the amateur group was significantly smaller than that of the professional group ($P < 0.01$). In the leg stirrup balance movement, the knee flexion/extension angle of the balancing leg in the amateur group was significantly greater than that in the professional group ($P < 0.01$), and the hip flexion/extension angle, hip inversion/abduction angle, knee flexion/extension moment, hip flexion/extension moment, the strength iliacus, gluteus maximus, and obturator internus were significantly smaller than those in the professional group ($P < 0.01$). The integral EMG of the biceps femoris of the support leg in the amateur group was significantly smaller than that of the professional group ($P < 0.01$). The integral EMG of the lateral femoral muscle of the balance leg was significantly smaller than that of the professional group ($P < 0.01$).

**Conclusion**. In this study, we found that the balancing leg of the balancing movement has a larger hip joint angle, the stirrup balancing knee joint angle is smaller, and the hip and knee joint moments are larger. This is related to joint activity and muscle activation, and amateurs should pay attention to increasing the range of motion of the hip joint and

decreasing the range of motion of the knee joint when practicing to better stimulate exercise of the lower limb joints. In addition, the practice of balancing movements should strengthen the iliacus muscle, which plays an important role in maintaining the stable balance of the lower limbs, and strengthen the knee flexor and extensor muscles and hip adductor/abductor muscles of the balancing leg, thus promoting the stability of the balancing leg movements.

## INTRODUCTION

Tai Chi is a multimodal exercise that combines aerobic exercise (*Tsoi et al., 2023*), resistance exercise (*Jones et al., 2023*), flexibility training, and neuromotor training (*Alsubiheen et al., 2017*); it exerts a comprehensive impact on the body's exercise effects and is widely used as a nonpharmacological preventive measure to prevent falls and improve balance in the elderly (*Wiedenmann et al., 2023*; *Yang et al., 2022*).

The stability of lower limb support is a major risk factor for falls (*Chew-Bullock et al., 2012*), and while early diagnosis and treatment of falls are important, prevention is even more important (*García-Pinillos et al., 2015*). In daily life, the risk of falls can be reduced if the stability of lower limb support is enhanced (*Pua et al., 2017*). A study by *Medeiros Barbosa et al. (2019)* showed that enhancing the stability of the lower limbs ensures smooth displacement of the body's center of gravity, thus maintaining the body's balance in the horizontal direction during exercise. The study by *Śliwowski et al. (2021)* noted that muscle strength is the key to lower limb support ability, strong muscle strength can make the lower limbs more stable during exercise. In addition, the study by *Sung & Kim (2018)* also proved that factors such as the range of motion and force on the joints of the lower limbs affect the balance of the limbs and that movement exercises can improve the flexibility of joint movements and enhance the stability of the limbs. However, Tai Chi has received extensive attention for its research in promoting human health. especially because its unique footwork and balancing movements play an important role in promoting balance (*Nyman et al., 2018*). The balancing movements of Tai Chi can improve lower limb proprioception (*Ye, Sun & Gao, 2017*) and enhance muscle strength (*Zhang et al., 2017*), thus promoting static and dynamic balance (*Penn et al., 2019*). The single-leg balancing movement in Tai Chi is difficult for amateurs, especially in the practice process; the instability of the supporting leg and balancing leg leads to the inability to complete the balancing movement, and it is easy for amateurs to cause sports injuries if they cannot perform the standardized practice (*Ke, 2022*).

At present, most of the research on Tai Chi worldwide focuses on intervention effects and physiological indicators, but there is little research on the movement mechanism and exercise mechanism of Tai Chi balance movements. In this article, the two most common balance movements in Tai Chi, "knee lift balance" and "leg stirrup balance", were selected for study. Based on computer simulation modeling, the lower limb biomechanical

characteristics of the two balancing movements were studied, and the lower limb kinematic and dynamic parameters of the supporting leg and balancing leg were analyzed with the aim of revealing the movement mechanisms of Tai Chi balancing movements and providing scientific theoretical guidance for amateurs.

## MATERIAL AND METHODS

### Participants

In this article, 16 male subjects were recruited and divided into two groups, one professional group and the other amateur group. The inclusion criteria for the professional group were national fitness level Tai Chi athletes and top three achievements in national Tai Chi competitions, and after recruitment screening, eight national team Tai Chi athletes were recruited, all from Beijing Sports University. The inclusion criteria for the amateur group were practitioners who had been practicing Tai Chi for 1 year. After recruitment screening, a total of eight Tai Chi practitioners were recruited, all from Beijing Normal University. Through participant physical status survey, all 16 subjects had no limb injury or surgical trauma within the last year, and all were right-side dominant with good health status. The subjects' basic physical parameters (age, height, weight, BMI, leg length, hip width, knee width, ankle width) were collected before the test, and there was no significant difference between the physical parameters of the professional and amateur groups. The lower limb bone muscle models of different subjects were established according to the body parameters of different subjects.

Prior to experimental testing, we have received informed consent from all participants. The subjects were asked to understand the purpose and process of this study, signed the informed consent form and passed the moral ethical review by the Ethics Committee of China National Rehabilitation Center (No. S20220206).

The sample size was small because there was a special group of subjects in this study, *i.e.,* national-level Tai Chi athletes, However, this study used Gpower 3.1 software and conducted *post-hoc* tests on the sample size, and the results showed that the test efficacy was 0.82 ($\beta = 0.82$ and $>0.8$). It has been shown that a test efficacy greater than 0.8 indicates that the sample size is reliable (*Shahid Saif Ali Baig & Jaisharma, 2022*), so the sample size of this study is reliable. In addition, this article mainly analyzes the typical balance movements of Tai Chi, and despite the differences between men and women in physical qualities, there are no differences in technical movement characteristics. Finally, only male subjects were selected for this study.

## METHODS

### Instrumentation

In this study, all motion data were collected by 8 high-precision infrared motion capture systems (BTS SMART DX 700; BTS Bioengineering, Lombardia, Italy), with parameters as follows: frequency of 250 Hz, resolution of 640 dpi × 480 dpi pixels, precision of 400 mm × 300 mm × 300 mm. Three 3D force plates (928E, Kistler, Switzerland) were used to measure the ground reaction force with a frequency of 1,000 Hz and a static detection

error of less than 0.5%. Surface EMG acquisition was selected using the BTS surface EMG test system to record the integrated EMG (iEMG) of the selected muscles within the test at a frequency of 1,000 Hz (*Li et al., 2022*). All instruments were synchronized for data acquisition through a synchronization system and connected to an experimental recorder to record the entire experiment.

## Typical balance movements of Tai Chi

This article studies two of the most typical balancing movements in Tai Chi: knee lift balance and leg stirrup balance (*National Sports College Textbook Committee, 2003*), which are frequently practiced by Tai Chi practitioners and represent the most typical balancing movements in Tai Chi (Fig. 1). The purpose of this article is to discuss the technical characteristics and exercise effects of the two typical balance movements of Tai Chi.

## AnyBody simulation

The musculoskeletal model of the lower limbs was built by the software AnyBody 7.2 (AnyBody Technology, Aalborg, Denmark), which processes 3D motion capture dynamics. The AnyBody 7.2 software has been validated by many experiments to operate with high reliability and accuracy (*Mubarrat & Chowdhury, 2023*; *Engelhardt et al., 2021*).

The musculoskeletal model built in AnyBody is a standard multibody dynamics model that consists of rigid components (*e.g.*, human skeleton or external objects), kinematic actuators (*e.g.*, body movements) and force/torque actuators (*e.g.*, muscles). The forces and torques in motion are simulated by multibody dynamics simulation. AnyBody software includes more than 1,000 muscle elements (*Damsgaard et al., 2006*) and allows analysis of the forces, deformations, elastic properties of muscle cusps, antagonistic muscle actions, and other properties useful to the human body for individual muscles, bones, and joints in the model (*Rasmussen, Boocock & Paul, 2012*; *Oral, Gönen & Özcan, 2017*). In this article, we study the motion data of typical balance movements of Tai Chi based on AnyBody's lower limb bone muscle model and establish the lower limb bone muscle model of typical balance movements of Tai Chi for the first time. After the optimization algorithm, the muscle synergy problem of the bone muscle model is solved, and the validity of the data of the model is further optimized. The aim is to scientifically conduct a comprehensive analysis of the typical balance movements of Tai Chi.

## Test protocol

The experiment was conducted in the motion biomechanics laboratory of Beijing Normal University. First, the BTS infrared motion capture cameras were adjusted in terms of position and height, with a distance of at least 2 m and a height of at least 3 m between each camera in a semiarc around the test center. Before the experiments, the equipment was adjusted to ensure that each camera could capture the complete movement of the subject and the whole body, including the calibration of the spatial global coordinate system and the calibration of the force table coordinates. The surface EMG system was turned on to interconnect with the IR motion system for surface EMG selection, and the VIXTA test recording camera was turned on to record the complete process of the experiment.

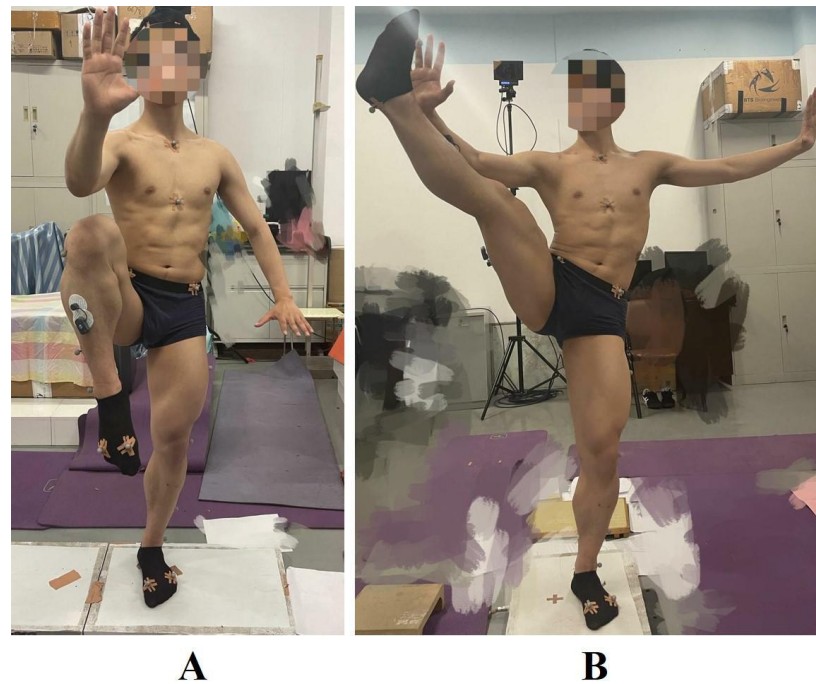

**Figure 1  Tai Chi in two typical balancing movements: (A) knee lift balance; (B) leg stirrup balance.**

Subjects wore uniform shorts and black lab socks, which in turn reduced the jitter from clothing and the error of different shoes on the force measurement table. Twenty-five motion-capture reflective dots were attached to the subjects at the bony marker points according to the requirements of the lower limb bone muscle model (*Li et al., 2023a*) (Fig. 2). Before MVC acquisition, the skin surface oil was wiped off with 75% alcohol cotton swabs, and the hair on the selected muscle belly was removed with a shaver to prevent interference with the surface MVC signal. Four surface muscles were selected: lateral femoral, biceps longus, tibialis anterior, and gastrocnemius, and the largest part of each muscle belly was selected. The surface electrode was applied to the muscle belly bulge of the tested muscle along its course and in the same direction as the muscle fiber. The MVC data were collected for each subject before the test (*Souza et al., 2021*), thus enabling the standardization of differences in muscle firing capacity across subjects (*Dankaerts et al., 2004*).

## Data collection and analysis

Since the dominant side of all subjects was the right side and the 2 typical balance movements of Tai Chi were symmetrical on the left and right sides, only the right-side balance movements were analyzed in this article, with the analysis focusing primarily on the movement characteristics of the right support leg and balance leg.

Participants started from a standing position and completed the motor movements of knee lift balancing and leg stirrup balancing on a force measuring board. Each movement was scheduled three times, and the average of the three data peaks was recorded. The

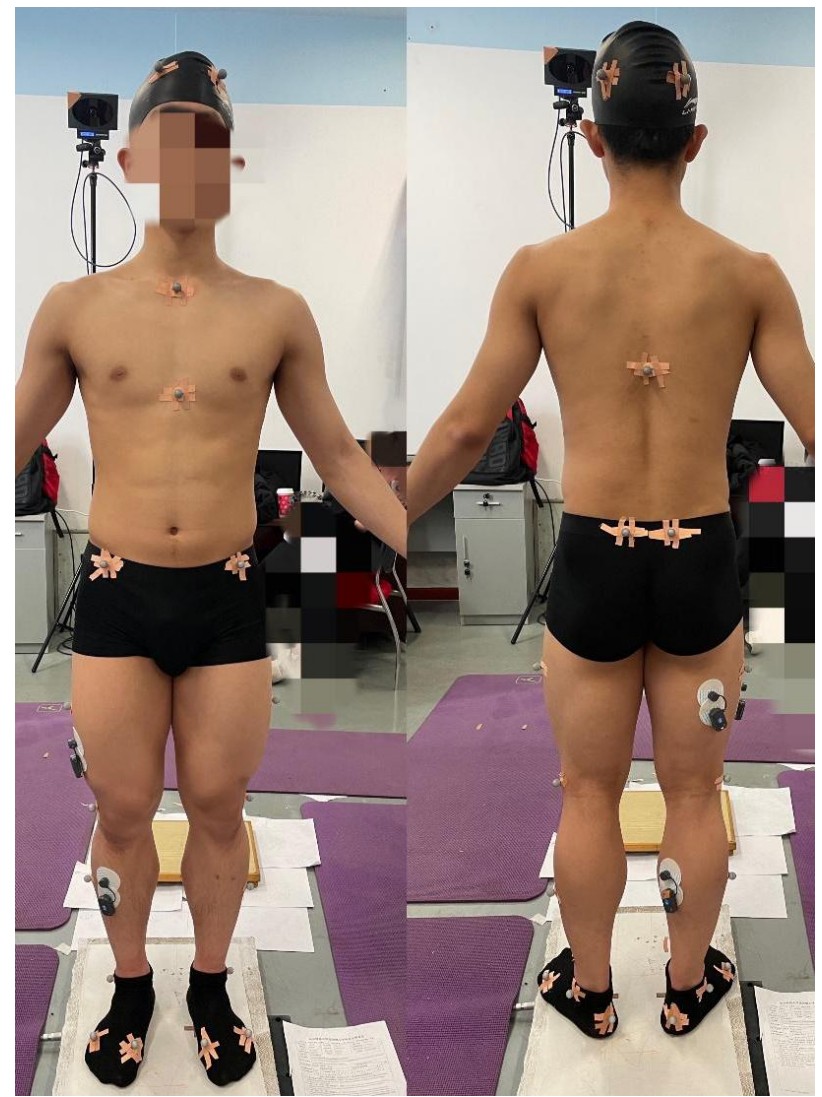

**Figure 2  Twenty-five marker points were attached to the bone marker points of the subject.**

three sets of movements were completed within 4–6 s. The data selection range was steady state after the movement was completed. All movements were selected with 800 frames of images.

## Kinematics and dynamics data

The BTS system acquisition software SMART Capture was used to collect data according to the lower limb motion model through the eight BTS SMART DX 700 cameras to model the movement data of the reflective markers on the subject. After the frame number interception of the subject data, the BTS tracker was used for model linkage, and the force platform data of different axial force calibrations was unified, processed, and exported to the C3D format file. According to the AnyBody human modeling simulation software, the
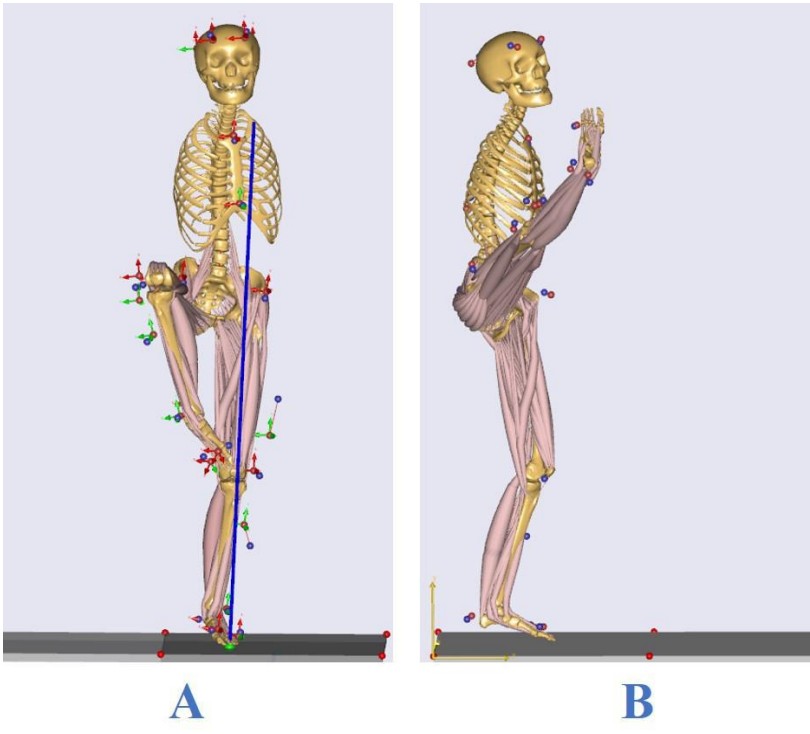

**A**          **B**

**Figure 3    Bone muscle models for two typical balance movements in Tai Chi: (A) knee lift balance; (B) leg stirrup balance.**

intercepted kinematic data C3D files were imported into the AnyBody 7 simulation software for calculation, and a bone and muscle simulation model of typical balance movements of Tai Chi was established (Fig. 3). The calculation flow adhered to the following order: reflex marker optimization—dynamics calculation—inverse dynamics calculation. The joint angle, joint moment, and muscle strength data for the exercise state were obtained after the calculation was completed, with the joint angle corresponding to the maximum peak state of movement completion and the kinetic index imported into Excel for normalization, with the joint moment divided by body weight (Nm/BW) (*Plaza-Manzano, 2023*) and the lower extremity muscle strength divided by body weight and multiplied by 100% (N/BW) (*Li et al., 2023b*). Standardized data are available for comparison between different groups.

## Surface EMG data

Data processing of surface EMG signal data was performed using SMART Analyzer software to filter and rectify the raw EMG data, and a filter range of 20–400 Hz was selected to remove low-frequency clutter and high-frequency spikes (*Clancy, Morin & Merletti, 2002*; *Brown, Brookham & Dickerson, 2010*), resulting in a smoother EMG signal. The EMG was corrected (*Špulák et al., 2014*), based on which the processed EMG data were analyzed in the time domain to derive the integral EMG values for each muscle. By measuring the MVC value of each muscle of the subject, normalization of the EMG signal was performed to eliminate the differences in the muscle firing capacity of different subjects, which in turn

enabled the comparison of the integral EMG values of different groups of subjects (*Reaz, Hussain & Mohd-Yasin, 2006*).

## Statistical analysis

Statistical analysis of the relevant indices was performed using SPSS 26.0 statistical software, and the data are expressed as the mean ± standard deviation (mean ± SD). Independent samples t tests were used to compare the lower limb movement data of the 2 typical Tai Chi balance movements, with $P < 0.01$ indicating a significant difference and statistical significance.

## RESULTS

### Comparison of lower limb joint angles for typical balance movements in Tai Chi

As shown in Table 1, in the knee lift balancing movement, the hip AB/AD (abduction/adduction) angle of the balance leg of the amateur group was significantly smaller than that of the professional group ($P < 0.01$), and there was no significant difference between the support leg and the professional group.

In the leg stirrup balance movement, the knee FL/EX (flexion/extension) angle of the balancing leg of the amateur group was significantly greater than that of the professional group ($P < 0.01$), and the hip FL/EX angle and hip AB/AD angle were significantly smaller than those of the professional group ($P < 0.01$).

### Comparison of lower limb joint torque for typical balance movements in Tai Chi

As shown in Table 2, the lower limb torque joint moments were compared. In the knee lift balancing movement, the hip FL/EX (flexion/extension) torque of the balancing leg in the amateur group was significantly less than that in the professional group ($P < 0.01$), and there was no significant difference in the lower limb triple joint moments of the supporting leg.

In the leg stirrup balance, the knee FL/EX and hip FL/EX torque of the balance leg of the amateur group were significantly less than those of the professional group ($P < 0.01$), and there was no significant difference with respect to the support leg *versus* the professional group.

### Comparison of lower limb muscle strength in typical balance movements of Tai Chi

As shown in Table 3, in the knee lift balancing movement, the iliopsoas muscle strength of the support leg in the amateur group was significantly lower than that of the professional group ($P < 0.01$), and the strength of the rectus femoris muscle, biceps femoris short capitis, and iliacus muscle of the balance leg was significantly lower than that of the professional group ($P < 0.01$).

In leg stirrup balance, the strength of the iliacus muscle of the support leg was significantly lower in the amateur group than in the professional group ($P < 0.01$), and the strength of

Li et al. (2023), *PeerJ*, DOI 10.7717/peerj.15817

**Table 1 Mean ± standard deviation (mean ± SD) of joint angles at the moment of maximum peak of lower limb joints for two typical balance movements.**

| Joint angle unit. (°) | Knee lift balance | | | | Leg stirrup balance | | | |
|---|---|---|---|---|---|---|---|---|
| | Support leg | | Balance leg | | Support leg | | Balance leg | |
| Joint | Amateur | Professional | Amateur | Professional | Amateur | Professional | Amateur | Professional |
| Ankle PF/DF | 22.65 ± 4.34 | 24.88 ± 3.15 | 28.69 ± 8.40 | 31.88 ± 7.51 | 18.22 ± 3.14 | 18.44 ± 3.42 | 14.60 ± 2.54 | 13.35 ± 3.93 |
| Knee FL/EX | 70.02 ± 21.70 | 81.92 ± 26.96 | 102.04 ± 16.09 | 104.40 ± 37.82 | 33.79 ± 11.57 | 28.19 ± 3.19 | 23.63 ± 1.20[*] | 12.48 ± 1.84 |
| Hip FL/EX | 51.99 ± 25.03 | 59.80 ± 27.08 | 69.06 ± 23.54 | 80.89 ± 22.46 | 10.03 ± 7.74 | 14.19 ± 10.24 | 58.09 ± 10.93[*] | 83.17 ± 10.90 |
| Hip AB/AD | 14.26 ± 8.38 | 15.97 ± 6.43 | 11.56 ± 4.62[*] | 23.17 ± 2.67 | 22.49 ± 5.79 | 25.71 ± 4.33 | 26.76 ± 7.39[*] | 37.89 ± 6.50 |
| Hip rotation | 16.41 ± 6.37 | 17.80 ± 5.81 | 16.28 ± 4.34 | 17.42 ± 6.63 | 18.39 ± 3.02 | 18.74 ± 2.27 | 9.44 ± 4.15 | 9.99 ± 3.33 |

**Notes.**

*$P < 0.01$ indicates the comparison between the professional and amateur groups.

PF/DF, plantar flexion/dorsiflexion; FL/EX, flexion/extension; AB/AD, abduction/adduction.

Li et al. (2023), *PeerJ*, DOI 10.7717/peerj.15817

**Table 2  Mean ± standard deviation (mean ± SD) of maximum torque of lower limb joints for two typical balance movements.**

| Joint torque unit. (Nm/BW) | Knee lift balance | | | | Leg stirrup balance | | | |
|---|---|---|---|---|---|---|---|---|
| | Support leg | | Balance leg | | Support leg | | Balance leg | |
| Joint | Amateur | Professional | Amateur | Professional | Amateur | Professional | Amateur | Professional |
| Ankle PF/DF | 0.65 ± 0.14 | 0.70 ± 0.37 | 0.39 ± 0.33 | 0.55 ± 0.51 | 0.41 ± 0.20 | 0.49 ± 0.37 | 0.20 ± 0.04 | 0.24 ± 0.03 |
| Knee FL/EX | 0.29 ± 0.20 | 0.31 ± 0.26 | 0.20 ± 0.15 | 0.30 ± 0.24 | 0.16 ± 0.01 | 0.20 ± 0.07 | 0.10 ± 0.02* | 1.11 ± 0.01 |
| Hip FL/EX | 0.33 ± 0.13 | 0.37 ± 0.13 | 0.37 ± 0.06* | 1.44 ± 0.14 | 0.05 ± 0.03 | 0.05 ± 0.08 | 0.49 ± 0.08* | 2.78 ± 0.05 |
| Hip AB/AD | 0.27 ± 0.11 | 0.29 ± 0.11 | 0.23 ± 0.20 | 0.29 ± 0.29 | 0.01 ± 0.01 | 0.01 ± 0.01 | 0.24 ± 0.10 | 0.32 ± 0.05 |
| Hip rotation | 0.09 ± 0.04 | 0.09 ± 0.03 | 0.10 ± 0.06 | 0.16 ± 0.15 | 0.06 ± 0.01 | 0.08 ± 0.02 | 0.14 ± 0.05 | 0.15 ± 0.06 |

**Notes.**

*$P < 0.01$ indicates the comparison between the professional and amateur groups.

PF/DF, plantar flexion/dorsiflexion; FL/EX, flexion/extension; AB/AD, abduction/adduction.

**Table 3** **Mean ± standard deviation (mean ± SD) of maximum muscle strength of lower limb muscles for two typical balance movements.**

| Muscle strength unit. (N/BW) | Knee lift balance | | | | Leg stirrup balance | | | |
|---|---|---|---|---|---|---|---|---|
| | Support leg | | Balance leg | | Support leg | | Balance leg | |
| Muscle | Amateur | Professional | Amateur | Professional | Amateur | Professional | Amateur | Professional |
| Soleus | 9.11 ± 2.23 | 9.26 ± 2.45 | 4.50 ± 1.61 | 4.23 ± 1.24 | 9.40 ± 1.03 | 10.00 ± 2.28 | 3.23 ± 0.34 | 3.62 ± 1.12 |
| Gastrocnemius lateralis | 4.04 ± 1.66 | 4.21 ± 1.27 | 1.86 ± 1.07 | 2.05 ± 1.31 | 0.02 ± 0.04 | 0.03 ± 0.02 | 0.19 ± 0.16 | 0.24 ± 0.20 |
| Gastrocnemius medial | 8.00 ± 2.97 | 8.64 ± 3.45 | 3.31 ± 1.75 | 3.67 ± 1.76 | 8.06 ± 3.12 | 8.55 ± 2.18 | 0.01 ± 0.01 | 0.03 ± 0.02 |
| Tibialis posterior | 1.32 ± 0.93 | 1.59 ± 0.38 | 1.92 ± 1.88 | 2.22 ± 1.86 | 8.34 ± 3.41 | 8.70 ± 2.62 | 0.35 ± 0.09 | 0.42 ± 0.13 |
| Tibialis anterior | 1.35 ± 1.09 | 1.78 ± 0.58 | 5.04 ± 1.83 | 5.07 ± 1.54 | 6.25 ± 2.03 | 6.69 ± 1.16 | 0.39 ± 0.25 | 0.42 ± 0.33 |
| Peroneus brevis | 1.90 ± 1.61 | 1.91 ± 0.98 | 1.05 ± 0.85 | 1.12 ± 0.69 | 0.16 ± 0.05 | 0.18 ± 0.02 | 0.22 ± 0.20 | 0.39 ± 0.17 |
| Peroneus longus | 3.63 ± 2.14 | 4.01 ± 1.12 | 8.80 ± 1.58 | 8.90 ± 2.88 | 0.37 ± 0.06 | 0.42 ± 0.08 | 0.47 ± 0.12 | 0.55 ± 0.32 |
| Lateral femoral muscle | 6.03 ± 2.29 | 6.65 ± 2.45 | 7.74 ± 3.14 | 7.84 ± 2.53 | 1.09 ± 0.41 | 1.23 ± 0.48 | 0.03 ± 0.04 | 0.06 ± 0.03 |
| Medial femoral Muscle | 2.98 ± 1.12 | 3.11 ± 1.20 | 2.52 ± 2.05 | 3.05 ± 1.08 | 1.00 ± 0.84 | 1.16 ± 0.49 | 0.02 ± 0.03 | 0.04 ± 0.02 |
| Middle femoral muscle | 0.72 ± 0.22 | 0.75 ± 0.27 | 1.86 ± 0.75 | 2.06 ± 0.32 | 0.50 ± 0.23 | 0.66 ± 0.38 | 0.01 ± 0.02 | 0.02 ± 0.01 |
| Rectus femoris muscle | 3.32 ± 1.44 | 3.19 ± 1.13 | 2.66 ± 1.24* | 6.13 ± 1.67 | 0.32 ± 0.04 | 0.36 ± 0.03 | 3.57 ± 0.83 | 3.51 ± 0.74 |
| Biceps femoris longus capitis | 3.37 ± 1.23 | 3.87 ± 1.69 | 1.51 ± 1.07 | 2.33 ± 1.18 | 0.51 ± 0.17 | 0.61 ± 0.28 | 0.29 ± 0.20 | 0.38 ± 0.07 |
| Biceps femoris short capitis | 0.75 ± 0.46 | 0.99 ± 0.74 | 0.70 ± 0.26* | 1.41 ± 0.43 | 0.25 ± 0.07 | 0.26 ± 0.05 | 1.01 ± 0.24 | 1.15 ± 0.43 |
| Sartorius | 1.28 ± 0.79 | 1.46 ± 0.90 | 1.77 ± 0.52 | 1.92 ± 0.59 | 0.58 ± 0.17 | 0.68 ± 0.44 | 2.45 ± 0.58 | 2.71 ± 1.13 |
| Iliacus muscle | 0.67 ± 0.25* | 2.30 ± 0.15 | 1.05 ± 0.73* | 3.76 ± 0.58 | 1.50 ± 0.18* | 3.67 ± 0.46 | 2.65 ± 0.53* | 4.57 ± 0.78 |
| Psoas major muscle | 1.08 ± 0.44 | 1.18 ± 0.69 | 1.50 ± 0.60 | 1.76 ± 0.64 | 1.11 ± 0.74 | 1.24 ± 0.09 | 1.76 ± 0.87 | 1.79 ± 0.92 |
| Gluteus maximus | 2.41 ± 0.43 | 2.57 ± 0.68 | 2.41 ± 1.56 | 2.84 ± 1.37 | 0.34 ± 0.38 | 0.40 ± 0.13 | 1.02 ± 0.78* | 2.51 ± 0.63 |
| Obturator externus | 0.94 ± 0.89 | 1.09 ± 0.87 | 1.39 ± 1.23 | 1.58 ± 1.40 | 3.48 ± 0.26 | 3.57 ± 0.13 | 0.44 ± 0.72 | 0.48 ± 0.25 |
| Obturator internus | 1.56 ± 0.54 | 1.64 ± 0.52 | 1.40 ± 0.93 | 1.44 ± 0.37 | 0.85 ± 0.07 | 0.91 ± 0.57 | 1.12 ± 0.68* | 3.98 ± 0.25 |

**Notes.**

*$P < 0.01$ indicates the comparison between the professional and amateur groups.

the balance leg, iliacus, gluteus maximus, and closed inner muscle was significantly lower in the amateur group than in the professional group ($P < 0.01$).

## Comparison of muscle integral EMG in the lower limbs of typical balance movements in Tai Chi

As shown in Table 4, there was no significant difference in the integral EMG values of the lower limb muscles between the amateur and professional groups in the knee lift balancing movement, and in the stirrup balancing, the integral EMG of the biceps femoris of the supporting leg was significantly smaller in the amateur group than in the professional group ($P < 0.01$). The integral EMG of the lateral femoral muscle of the balancing leg was significantly smaller than that of the professional group ($P < 0.01$).

# DISCUSSION

## Analysis of lower limb joint angles for typical balance movements in Tai Chi

The joint angle of motion determines the extent of joint range of motion and reflects the accuracy of posture in the state of motion  (*Steinberg et al., 2006*). *Ashtiani, Azghani & Parnianpour*'s (*2018*) study noted that correct lower limb joint angles have a positive effect on maintaining the stability of limb motion. In this article, by analyzing two typical Tai Chi balancing movements, knee lift balance and leg stirrup balance, we found that the hip joint AB/AD angle of the balancing leg was greater in the professional group, which indicates that the hip joint AB/AD angle has a great influence on the balancing leg in the practice of knee lift balance and leg stirrup balance Tai Chi balancing movements. Both knee lift balance and leg stirrup balance are single leg support leg lift balance movements, and their movements require the balance leg to be lifted above the thigh level, thus showing the hip joint as the main source of movement: however, practicing amateurs mainly focus on the hip joint FL/EX movement and ignore the hip joint AB/AD movement. These two types of Tai Chi balancing movements require that after the hip joint FL/EX movement exceeds the horizontal position, the hip joint needs to move inward to increase the range of motion of the balancing leg inward, thus preventing the limb from falling like the rear lateral side due to excessive hip joint FL/EX movement of the balancing leg (*Kim & Kang, 2020*; *Catena et al., 2019*). This is consistent with the findings of *Francis, Gray & Perrem (2018)* because performing the single-legged leg lift movement is a hip-joint-dominated movement, so when increasing the height of the balancing leg, it is necessary to slightly increase the lateral joint range of motion of the hip joint and increase the AB/AD joint range of motion to ensure the stability of the balancing leg. Therefore, in knee lift balance and leg stirrup balance movements, the hip joint AB/AD angle of the balance leg is greater in the professional group, while amateurs often find that the hip joint AB/AD angle of the balance leg is smaller when performing knee lift balance and leg stirrup balance movements. This is detrimental to the stability of the lower limbs. In addition, during leg stirrup balance, the hip joint FL/EX angle of the balance leg in the professional group was greater than that in the amateur group, while the knee joint FL/EX angle in the amateur group was significantly greater than that in the professional group. This is because leg stirrup balancing belongs

Li et al. (2023), *PeerJ*, DOI 10.7717/peerj.15817

**Table 4  Mean ± standard deviation (mean ± SD) of maximum integrated EMG of lower limb muscles for two typical balance movements.**

| iEMG unit. (uV) | Knee lift balance | | | | Leg stirrup balance | | | |
|---|---|---|---|---|---|---|---|---|
| | Support leg | | Balance leg | | Support leg | | Balance leg | |
| Muscle | Amateur | Professional | Amateur | Professional | Amateur | Professional | Amateur | Professional |
| Lateral femoral muscle | 170.13 ± 6.24 | 172.34 ± 7.6 | 187.53 ± 14.61 | 196.9 ± 13.21 | 145.08 ± 4.76 | 146.21 ± 3.14 | 152.43 ± 1.56[*] | 188.13 ± 2.71 |
| Biceps femoris | 84.41 ± 10.14 | 87.98 ± 13.25 | 105.5 ± 7.62 | 107.18 ± 5.73 | 307.79 ± 6.24[*] | 378.15 ± 7.11 | 413.62 ± 5.66 | 416.29 ± 4.38 |
| Tibialis anterior muscle | 94.35 ± 4.32 | 95.61 ± 6.17 | 137.24 ± 7.32 | 139.96 ± 5.62 | 209.13 ± 8.21 | 212.91 ± 5.64 | 231.4 ± 9.09 | 236.17 ± 12.67 |
| Gastrocnemius muscle | 99.45 ± 9.45 | 102.01 ± 10.52 | 103.44 ± 10.04 | 105.73 ± 9.51 | 194.6 ± 2.14 | 196.43 ± 3.06 | 201.43 ± 10.11 | 206.15 ± 7.93 |

**Notes.**

*$P < 0.01$ indicates the comparison between the professional and amateur groups.

to the balancing action of extending the knee and lifting the leg, and the greater the hip flexion and extension angle, the greater the range of motion of hip flexion and extension: this shows that the balancing leg of the professional group lifts higher, but the balancing leg of the amateur group cannot lift higher when it is lifted upward, which has much to do with the hip flexion and extension flexibility and muscle strength of the amateur group of Tai Chi. For this group, when straightening the knee, the knee flexion and extension muscles are weak, resulting in the inability to complete the movement in a standardized manner, which ultimately manifests itself in the amateur group through a larger knee flexion and extension angle and the wrong leg bending movement of stirrup balance. The flexibility and strength of hip flexion and extension can be improved by this unique training method of leg stirrup balance (*Hrysomallis, 2009*). In the *Leppanen (2020)* study of knee extension and leg raising exercises, it was found that increasing the hip flexion and extension angles of motion effectively enhanced the muscle recruitment of hip flexion and extension, as well as effectively promoting joint proprioception in the lower extremity. The study by *Khalaj, Vicenzino & Smith (2021)* also noted that hip flexion and extension, as well as knee muscle groups, could enhance lower extremity proprioception and improve lower extremity stability. This can also improve balance and stability of the lower limbs. When practicing leg stirrup balance, amateurs should increase the hip flexion and extension movement angle and decrease the knee flexion and extension movement angle.

## Analysis of the forces on the joints of the lower limbs in typical balance movements of Tai Chi

Joint torque is the force generated by the change in the relative positions of joints and muscles during movement (*Yang et al., 2019*). It is the force produced by a joint in different states of motion and can reflect the state of strength of the joint and muscle in motion (*Mausehund & Krosshaug, 2021*). *Smith et al. (2021)* showed that the greater the joint torque during motion, the greater the strength of the corresponding muscle group. This rise in joint torque during exercise not only reflects the state of movement but also provides a positive stimulus to exercise the joints. In this article, we found that the hip flexion-extension torque of the balance leg was significantly greater in the professional group in both knee lift balance and leg stirrup balance movements, which may be related to the lifting height of the balance leg, and through observation, we found that the range of motion of the hip flexion-extension of the balance leg was greater when performing these two typical balance movements, and therefore, the height of the upward lifting of the balance leg was higher in the professional group. *Moreira et al. (2021)* also confirmed that increased range of motion of the joint causes the joint torque to rise, which is the reason for the increased flexion and extension moment of the hip joint in the professional group. Tai Chi balancing movements can effectively stimulate the hip joint and improve joint proprioception and coordination of the surrounding muscles by increasing the range of motion of the joint and raising the joint torque. A study by *Eustace (2022)* quantified joint torque during exercise and found that an increase in joint torque during exercise can bring positive stimulation to the joint and improve joint control of the limb, so the hip joint can be stimulated by the typical balancing movements of Tai Chi. The hip joint is the largest

moving part of the lower limb and the most important moving part in daily life, and its motility is important for maintaining the stability and balance of the lower limb (*De Blaiser et al., 2021*). When practicing Tai Chi balance movements, amateurs must pay attention to the height of hip flexion and extension and keep practicing the height of the leg lift to stimulate the exercise of the hip joint. This is very important for the training of lower limb stability. In addition, this article also found that the knee FL/EX angles were smaller in the professional group during leg stirrup balancing, but the knee FL/EX torque was higher. This is because the joint moment is positively correlated with the range of motion of the joint, but also with the relative position of the muscles during exercise, and the strength of muscle activation also exerts an important effect on the joint torque. In leg stirrup balance, the balancing leg requires the knee joint to be straight, which requires the knee extensor muscles to control the balancing leg to keep it straight. However, the amateur has a greater knee flexion and extension angle because the knee muscles are not strong enough, which results in a bent leg relaxation state and less knee flexion and extension torque. Thus, the stimulation effect on the knee joint is not sufficient to meet the requirements of the movement. The professional group can better stimulate the knee joint by intensifying the training of straightening the balance leg, which shows a greater joint torque during balance leg movement. *Sadeghi et al. (2021)* study concluded that the knee joint plays a supporting role in lower limb movements, and with the support ability of the knee joint, the balance of the lower limb can be better maintained. Therefore, the balancing movements in Tai Chi positively impact the training of the supporting ability of the knee joint, and when practicing leg stirrup balancing, attention should be devoted to the straightening of the balancing leg to obtain better training effects.

## Analysis of lower limb muscle strength and recruitment ability of typical balance movements in Tai Chi

The lower limb muscle force activation states of the two balance movements of Tai Chi were analyzed, and the recruitment ability of the lower limb muscles was observed through the integral EMG values of surface EMG. We found that the muscle strength of the iliacus muscle of the supporting leg and the rectus femoris, the biceps femoris short capitis, and the iliacus muscle of the balancing leg was greater in the professional group when performing the knee lift balancing movement, while the muscle strength of the iliacus muscle of the supporting leg and the iliacus muscle of the balancing leg, the gluteus maximus, and obturator internus was greater in the professional group in leg stirrup balancing. The results showed that the iliacus muscle plays a vital role in the balancing movement of Tai Chi, which indicates that the strength of the iliacus muscle plays a vital role in the balancing movement. *Alam et al. (2019)* reported that the iliacus muscle is the main muscle that controls the balance and stability of the lower limbs, and that enhancing the muscle strength of the iliacus muscle can effectively promote the core strength to enhance the balance of the limbs, which is consistent with the results of this article. The professional group showed a significant increase in the strength of the iliacus muscle during the balancing movements, which also proves the significance of the iliacus muscle to the balancing movements. The amateurs of Tai Chi should strengthen the iliacus muscle during

balancing movement training. When training for balance movements, the muscle strength of the iliacus muscle should be strengthened so that the balance movements of Tai Chi can be better performed. In addition, the two balancing movements differ in the activation of the muscle groups that balance the leg. The knee lift balancing movement is mainly performed by the knee flexors/extensors (rectus femoris muscle, biceps femoris short capitis) as the main muscle groups of exertion. In contrast, the muscle groups that exert force in leg stirrup balancing are the hip adductors/extensors (gluteus maximus, obturator internus). This is because the balance leg of the knee lift balance is a flexed knee leg lift balance. *Yuan et al. (2019)* shows that knee lifts can effectively exercise the muscles around the knee joint and strengthen the knee flexors and extensors, thus improving the stability of the knee joint during exercise. The training of leg stirrup balance is important to exercise the hip adduction/abduction muscle groups. The hip adductor/abductor muscles are the main muscle groups that control the movement of the lower limb during sports (*Liu et al., 2021*), and improving their strength can enhance the ability of the hip joint to control the stability of lower limb movements (*Siriphorn & Chamonchant, 2015*). Although the activation and exercise of lower limb muscles vary among different Tai Chi balancing movements, all Tai Chi balancing movements can enhance the strength of lower limb muscles and promote lower limb balance and stability. By integrating electromyographic values, this study also found that in leg stirrup balancing, the professional group had stronger biceps femoris muscle recruitment in the supporting leg and stronger lateral femoral muscle recruitment in the balancing leg. This is because during the leg stirrup balance movement, the balance leg is prone to backward instability after lifting and requires the antagonistic force of the posterior thigh muscles of the supporting leg to ensure the stability of the body, so the biceps femoris muscle of the supporting leg has a higher recruitment capacity during the movement, while the anterior thigh muscles mainly drive the balance leg when it is lifted upward, and thus the lateral femoris muscle of the balance leg has a higher recruitment capacity when the leg is lifted upward. For amateurs in Tai Chi, the exercise of the biceps femoris and lateralis femoris should be strengthened to make the supporting leg more stable when balancing on the ground and to ensure the height of the balancing leg.

Strengths and limitations: This study is very meaningful to the Tai Chi movement. This article is the first scientific analysis of the typical balance movements of Tai Chi and establishes a bone muscle model of the lower limb movement for the balance movements of Tai Chi, which provides theoretical guidance for the scientification of Tai Chi. The study still has some limitations: as it was conducted to establish the lower limb bone muscle model of Tai Chi balance movements with a special population, the sample size is small, and the lower limb joint coordination pattern is not explored. The study will be continuously improved in future work. We will continue to explore scientific research on the Tai Chi movement.

## CONCLUSION

The study in this article draws the following conclusions. In the two typical balancing movements of Tai Chi, the hip angle, joint forces, and range of motion of the hip joint were significantly greater in the professional group than in the amateur group, which can promote the exercise effect of the joint. In addition, iliopsoas strength is important in both balancing movements and has a role in stabilizing postural stability. For beginners, strengthening the iliacus muscle strength training helps to better perform the balancing movements of Tai Chi. Overall, when practicing balancing movements, for amateurs the focus should be on increasing the range of motion of the hip joint and decreasing the range of motion of the knee joint to ensure better practice results. Strengthening the strength training of the iliopsoas, knee flexors and knee flexors and hip adductors/abductors can promote the training effect of balancing movements.

## ACKNOWLEDGEMENTS

The authors thank sports biomechanics laboratory of the Beijing Normal University for their support in data collection.

### Funding
The authors received no funding for this work.

### Competing Interests
The authors declare there are no competing interests.

### Author Contributions
- Haojie Li conceived and designed the experiments, analyzed the data, prepared figures and/or tables, authored or reviewed drafts of the article, and approved the final draft.
- Xin Wang performed the experiments, analyzed the data, prepared figures and/or tables, authored or reviewed drafts of the article, and approved the final draft.
- Zhihao Du conceived and designed the experiments, performed the experiments, analyzed the data, prepared figures and/or tables, and approved the final draft.
- Shunze Shen conceived and designed the experiments, performed the experiments, prepared figures and/or tables, and approved the final draft.

### Human Ethics
The following information was supplied relating to ethical approvals (*i.e.*, approving body and any reference numbers):

This study was reviewed and approved by the Ethics Committee of the China National Rehabilitation Center (No. S20220206).

### Data Availability
The data is available at Figshare: Li, Haojie (2023). Raw data of typical balance movements of Tai Chi. figshare. Dataset. https://doi.org/10.6084/m9.figshare.22574185.v1.

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
