# Peer review of "Analysis of technical characteristics of typical lower limb balance movements in Tai Chi: a cross-sectional study based on AnyBody bone muscle modeling"

_PeerJ, doi:10.7717/peerj.15817_

## Round 0.1 · original submission · Major Revisions

Dear Authors,

The reviewers and I have completed our evaluation of your manuscript and recommend a major revision before re-submission.

Please review the comments and resubmit your revised manuscript.

·

Basic reporting

Writing should be more straightforward and more accessible for a reader to understand. There needs to be more academic style of writing. An English language expert should be consulted to improve the manuscript related to these points.

Experimental design

Statistical analysis – Please mind that statement is incorrect when reporting both significance levels (p<0.05 and p<0.01) as the first value to be insignificant and the second to be significant, respectively. This is an error. The level of significance is determined as only one p-value. Therefore, authors should determine this value and re-run the analysis according to either p<0.05 or p<0.01.

Validity of the findings

Conclusion – In this part, the clear take-home message and the main results should be presented, keeping the conclusion simple and clear enough.

Additional comments

Ln50 "lower limb leg" – Please check if this is correctly written and rephrase if needed.
Ln58 "Tai Chi is" - The sentence starts identically as the previous one. Please consider rephrasing it.
Ln66,67 "strong leg strength" – What is the meaning of this? Please recheck this statement.
Ln71,72 “both at home and abroad” - What is meant by abroad here? The point is not clear. Please rephrase this sentence.

Ln88 “participants. the” - Upper case letter is used when starting a new sentence.

Ln91-93 - Please consider rephrasing this sentence.

Ln94 “although there are” - despite the.

Ln97 - "Participants" subsection should start with this sentence... Please correct accordingly.

Ln100,101 “The inclusion criteria for the amateur group were enthusiasts who had practiced Tai Chi for more than 1 year, and 8 students from Beijing…” - This is unclear. Please, rephrase this accordingly.

Ln102 “after investigation” - After what investigation precisely?

Ln201,202 – “with P < 0.05 indicating a difference and P < 0.01 indicating a significant difference and statistical significance.” - This seems to be confusing and unclear. Statistical significance is set either at p<0.05 or p<0.01, not both. Please correct this statement.

Ln207 “AB/AD” - What is the meaning of this abbreviation? Write it in the bracket, please.

Ln262 “FL/EX” - The meaning of this abbreviation is missing.

Ln265 “abduction/adduction” - Please use either abbreviation, as you used earlier in the text.

Ln311 "the professional balance leg group." – What is meant by this? Recheck the entire sentence and rewrite it if needed.

Ln341 “completed” – performed?

Ln346 “exercise can effectively exercise the muscles around” - Please rephrase this part of the sentence.

Ln374 “is larger“ - Larger compared to what? Please rewrite the sentence in a way it makes sense and improves the readability.

Ln382 "We" – We? Please recheck the statement's grammar and meaning (point) and rephrase.

·

Basic reporting

Dear Authors,
thank you for submitting the manuscript "Analysis of technical characteristics of typical lower limb
balance movements in Tai Chi: a cross-sectional study based on AnyBody bone muscle modelling". In my opinion, it is fascinating and well-written. The English used is clear and unambiguous, and sufficient and relevant background has been provided. Also, the article's structure appeared professional, with relevant results to hypotheses. Basically, I'd just suggest you synthesise your abstract, especially in results.

Experimental design

The research questions are well-defined, relevant and meaningful. Rigorous investigations have been performed to high technical and ethical standards. Methods have been described correctly. Although the authors have clearly stated that the sample size is a limitation and why it happened (lines 91-95), I'd suggest you estimate the statistical power achieved with your study design characteristics. This is my only minor suggestion related to your methods.

Validity of the findings

The findings are well-debated and compared with previous results. All underlying data have been provided and appeared controlled and robust. Just a suggestion on the statistical power has been provided above. The conclusions are well stated, linked to the original research question and limited to supporting results.

---

## Round 0.2 · accepted · Accept

Your manuscript has been accepted for publication. Congratulations!

·

Basic reporting

The manuscript has been significantly improved.

Experimental design

The experimental design meets the requirements of the journal.

Validity of the findings

The manuscript is significant for readers.

Additional comments

The manuscript has been significantly improved. I think it should be published.

·

Basic reporting

Clear and professional English has been used. Background with sufficient and relevant content has been provided. Professional structure. figures and tables have been presented.

Experimental design

The research questions are well-defined and relevant. Rigorous investigations have been performed.

Validity of the findings

All underlying data have been provided; they are robust, statistically sound, and controlled. Conclusions have been well stated, and linked to the original research question.

Additional comments

The authors revised all suggested comments.